# Stand-Biased Desks Impact on Cognition in Elementary Students Using a Within-Classroom Crossover Design

**DOI:** 10.3390/ijerph19095684

**Published:** 2022-05-07

**Authors:** Alexander L. Wallace, Ann M. Swartz, Chi C. Cho, Christine M. Kaiver, Ryan M. Sullivan, Krista M. Lisdahl

**Affiliations:** 1Department of Psychology, University of Wisconsin-Milwaukee, Milwaukee, WI 53211, USA; cmkaiver@uwm.edu (C.M.K.); rmsul@uwm.edu (R.M.S.); 2Department of Kinesiology, University of Wisconsin-Milwaukee, Milwaukee, WI 53211, USA; aswartz@uwm.edu; 3Center for Aging and Translational Research, University of Wisconsin-Milwaukee, Milwaukee, WI 53211, USA; chocc@uwm.edu

**Keywords:** cognitive function, standing, children, school, sedentary behavior, development

## Abstract

**Background:** There is emerging literature that standing desk interventions may help to improve cognitive performance in school-aged children. The current study examines how desks that promote standing affect cognition over the course of a school year in third, fourth, and sixth graders. **Methods:** Nighty-nine students between the ages of 8 and 12 (M = 10.23; 58% Male) were assigned to either stand-biased desks or traditional sitting desks. A within-classroom design was used with students switching desks after 9 weeks. Cognitive assessments and teacher behavioral ratings were administered at baseline and readministered before students switched desks and at the conclusion of the study. **Results:** There were no significant effects on cognition or behavioral ratings from standing-biased desk intervention. Grade significantly moderated the relationship between stand-biased desks and cognition in that third graders showed increased cognitive control (*p* = 0.02, f^2^ = 0.06). Further, sex moderated the relationship in that females at stand-biased desks showed increased cognitive control (*p* = 0.03, f^2^ = 0.04). **Conclusions:** These results suggest that stand-biased desks impact cognition depending on grade and sex, indicating a complex relationship that should be teased out further in future research. Stand-biased desks showed moderate improvements in cognition and no deleterious effects, suggesting that they may be a helpful classroom intervention for children in elementary school.

## 1. Introduction

Approximately half of U.S. children engage in over two hours of sedentary behavior a day [1]. Sedentary behavior (i.e., low energy expenditures) while in a sitting or reclining position [2] in children is largely associated with reduced physical health, self-esteem, and prosocial behaviors; however, research guidelines suggest any reduction in these sedentary behaviors can lead to improvements within these domains [3]. Furthermore, preliminary studies suggest that increased physical activity in school-age children improves cognition and academic achievement [4,5,6]. Additionally, the impact of movement seems promising for brain development, functional brain activity, and scholastic performance [6,7].

Despite the known benefits of movement and activity during this period of development, the nature of an academic classroom setting, i.e., primarily sitting at a desk [8], promotes these sedentary behaviors. To address these concerns, innovative ways to engage children in movement-based behavior while in the classroom setting have been important for reducing overall sedentary behavior [9,10]. Thus, the integration of physical activity within classrooms has shown promising improvements in cognitive functioning and academic achievement [11]; however, there is still much room for improvement. A major movement within classrooms to try and address these areas of potential improvements is the integration of standing desks within the classroom [12].

Standing desks represent a feasible and easily incorporable method for increasing movement within the classroom [13]. Studies utilizing standing desks have shown benefits in decreased sedentary behavior [14] and higher energy expenditures among school-aged children [15]. Further, some evidence has shown that standing-based desks may help to improve cognition within the classroom; preliminary results have shown significant improvements in academic engagement [16] and working memory in primary students [17] and set-shifting in secondary students [18]. These findings may be best explained through low-level physical activity that is achieved through standing desk use (i.e., increased energy expenditures), as standing desks have been shown to increase standing time and therefore increase low levels of activity through standing [19,20]. Although physical activity through standing is minimal [21], these low levels of activity may still have a positive impact. Further, while these effects on cognition are subtle, there were no deleterious cognitive, academic, or health-based effects in students using standing desks, further demonstrating that the utilization of these could be a low-risk classroom intervention.

Despite these promising preliminary studies, only one study known to date has examined the impact of standing desks on neurocognition in primary school students [17]. While this study is an important foundational step within the literature, it only measured a small range of cognition by utilizing two executive functioning measures (digit span and flanker task). Moreover, due to the low sample size and failure to account for within and between group baseline differences, further strides are needed to examine the effects of standing desks on neurocognition in primary students.

Neuropsychological functioning, specifically executive functioning, within developing children is often tied closely to externalizing problem behaviors (e.g., hyperactivity, attention problems) [22,23]. Research suggests that externalizing disorders have been related to wide variability in energy expenditure among children [24]. With standing desks leading to increased energy expenditure in primary students [15,16], it is possible that energy expenditure from utilizing standing desks may lead to a decrease in externalizing behaviors and greater executive control within the classroom. As previous research with this sample has shown that standing desks mitigated increases in sedentary behavior compared to their traditional desk using peers [14], it is reasonable to examine if these behavioral changes were captured by teacher rating forms as well. 

By using stand-biased desks versus sit-biased desks in a within-subject classroom crossover study design, we hypothesize that stand-biased desks will be associated with superior cognition and teacher ratings of executive functioning behaviors within third-, fourth-, and sixth-grade students. Specifically, we expect that children who have used a stand-biased desk for 9 weeks will experience improvements in cognitive domains of attention and working memory, as well as in-class teacher ratings of global executive functioning compared to students within the same classrooms who were not exposed to a stand-biased desk during the same period. Further, we will examine whether grade level or sex moderate these findings. 

## 2. Materials and Methods

### 2.1. Participants

Nighty-nine students in 3rd, 4th, and 6th grade (thirty-five 3rd graders, twenty-three 4th graders, and forty-one 6th graders) enrolled in an urban elementary school were recruited to be a part of this study. Participants were between the ages of 8 and 12 and predominately male (58.1% Male).

### 2.2. Procedures

Prior to the start of the study, research assistants attended classroom orientations to describe the study with parents and students and obtain consent and assent from interested families. Regardless of participation in the study, all students were assigned to a stand-biased or sitting desk (50%/50%) by each classroom teacher. Participants assigned to a sitting desk in the fall switched to a stand-biased desk after 9 weeks mid-winter and used a stand-biased desk for 9 weeks in the spring (Group 1). Vice versa, participants who were assigned to a stand-biased desk utilized it for 9 weeks and then switched to a sitting desk mid-winter for 9 weeks of use in the Spring (Group 2). A within-classroom crossover design was used to test the effects of desk intervention on cognition while preventing confounding effects of different classrooms and teaching environments. Measurements of cognitive performance occurred every 9 weeks and were obtained at baseline (September), mid-winter (December), and conclusion of the study (April) for all participants within a one-week timeframe. For more information on the study protocol, see Swartz, Tokarek, Lisdahl, Maeda, Strath, and Cho [14].

### 2.3. Materials

#### Standing-Biased Desks

Standing-biased desks were adjusted to a level where students could work from the desks comfortably while standing. All standing-biased desks were accompanied by stools for the option of sitting if students so wished. There were enough standing-biased desks in each classroom for approximately 50% of the students to have one each semester.

### 2.4. Outcome Measures

#### 2.4.1. NIH-Toolbox

The NIH-Toolbox was designed to assess important areas of cognitive functioning that are often impacted by health [25] as well as been shown to be reliable and valid for neurocognitive development in children [26]. The NIH-Toolbox is administered via iPad and contains a range of different cognitive tasks developed for and normed on participants aged 3–85 years. Age corrected scores were later utilized for analyses, with higher scores representing better performance. The tasks used for this study are listed below.

(1)Flanker. The Flanker task requires participants to indicate whether a center arrow is pointing left or right amongst a line of arrows that are either congruent (all pointing the same way as a central arrow) or incongruent (all pointing the opposite way of the central arrow). In this way, the Flanker task measures a participant’s selective attention and cognitive control.(2)List Sorting. The List Sorting task requires participants to remember a list of animals, foods, or both foods and animals. Participants were then asked to report the list in size order, with two category lists being reported one category at a time. In this way, the List Sorting task measures participants working memory.(3)Pattern Comparison. The pattern comparison task requires participants to report if two stimuli are the same or different and is a good measure of participants’ processing speed.(4)Picture Sequence Memory. In Picture Sequence Memory, participants are presented with a series of images while being told a story. The participants are then required to arrange the images in the serial order of what they were shown. In this way, participants are demonstrating skills of visual working memory and episodic memory.

#### 2.4.2. Teacher Report

The Behavior Rating Inventory on Executive Function, second edition (BRIEF-II), was used to assess teacher perceptions of executive functioning within the school environment [27]. The BRIEF-II provides reliable and valid data on children’s executive functioning skills in domains of Inhibition, Self-Monitoring, Shifting, Emotional Control, Initiation, Working Memory, Planning/Organization, Task Monitoring, and Organization of Materials [28]. These sub-scales make up domains of the child’s Behavioral Regulation, Emotional Regulation, and Cognitive Regulation as well as their overall Global Executive Functioning. Higher scores across all domains indicate higher executive dysfunction. 

### 2.5. Statistical Analyses

Baseline differences in demographic data between the sitting-to-standing group and the standing-to-sitting group were carried out with t-tests and chi-squares. Mixed effect models with random intercept utilizing a maximum likelihood approach were used to examine the effect of desk type, grade, gender, and their interactions on NIH toolbox and BRIEF outcomes. A dummy variable indicating the sequence of the intervention delivery was included in the model to control for the crossover design. All statistics were calculated using SAS (version 9.4; Cary, NC, USA). Cohen’s f-squared was computed for effect sizes using PROC MIXED [29]. All statistical decisions were made at *p* < 0.05. Further, effect sizes were interpreted using Cohen (1992) guidelines, f-square ≥0.02 = small, ≥ 0.05 = medium and ≥0.35 = large effects [30].

## 3. Results

### 3.1. Demographics

There were no significant differences between the sitting-to-standing group and the standing-to-sitting group by age (*p* = 0.72), sex (*p* = 0.84), race (*p* = 0.42), nor grade (*p* = 0.69). See Table 1 for more demographic information. 

### 3.2. NIH Toolbox

Participants did not significantly differ in NIH toolbox performance by standing desk group at baseline (see Table 2). Standing desks did not show statistically significant improvements in the NIH toolbox compared to traditional desks in the whole sample (*p*’s > 0.05).

#### 3.2.1. Grade

There was a significant desk-by-grade interaction in explaining differences in Flanker performance. Specifically, when using standing desks, third graders showed significant improvements on the Flanker task (β = 3.65, SE = 1.51, *p* = 0.02, f^2^ = 0.06; see Figure 1; Table 3), while fourth and sixth graders did not. Further, there was a significant interaction between time and grade level, with sixth graders showing significantly greater improvement on Pattern Comparison from the fall to the spring semester compared to third- and fourth-grade students, regardless of desk type (β = 6.64, SE = 2.64, *p* = 0.009, f^2^ = 0.13). 

#### 3.2.2. Sex

Sex significantly moderated the effects of standing desks on Flanker performance, with females in stand-biased desks demonstrating increased Flanker performance compared to males in stand-biased desks (β = 2.54, SE = 1.18, *p* = 0.03, f^2^ = 0.04; see Figure 2; Table 3).

### 3.3. BRIEF-II

Participants did not significantly differ on composite Teacher BRIEF-II scores at baseline. Standing desk groups did not have any significant effects on BRIEF-II scores.

## 4. Discussion

The current study aimed to investigate if stand-biased desks would improve cognition and executive functioning skills in elementary school students. Standing-biased desks were shown to be beneficial within a particular cognitive domain based on participant’s grade and sex. Specifically, third graders and females using standing-biased desks displayed benefits in selective attention and cognitive control. Standing-biased desks did not show any significant effect on teacher reports of executive functioning.

These findings somewhat differ from previous research that indicated that standing desks only showed improvements in working memory in elementary students, with no significant findings related to the flanker task [17]. However, our study was able to examine how grade acted as a moderator on the impact of standing desks and cognition. In looking at these factors, interesting patterns revealed that stand-biased desks influence differing domains of cognition depending on what stage of development they are introduced to classroom children. Third graders experienced significant improvements in flanker performance compared to their older counterparts, suggesting that standing desks may be important in improving selective attention and cognitive control at earlier stages of development compared to later stages. This might explain why studies examining selective attention with the flanker task in high-school freshmen saw no changes in task performance after using standing tasks [18]. 

As childhood and adolescence continue to be times of great brain development, the brain is more susceptible to external influences that may impact cognition [31]. Greater levels of physical activity and movement may improve brain health and downstream neurocognition. Indeed, increased physical activity (i.e., moderate-to-vigorous activity) has been implicated in affecting the course of cognitive development [32]. Links between exercise and brain-derived neurotrophic factors have been proposed as a potential mechanism behind changes from increased physical activity in brain structures, functioning, and cognition [33]. These markers have been an important measure for the changes in cognition [34] and neurogenesis within animal models [35]. Stand-biased desks (representing low-level physical activity) may, in turn, relate to subtle increases in cognition through the aforementioned mechanisms. As previously reported in this sample, individuals at stand-biased desks engaged in less sedentary behavior compared to children at traditional desks [14]. Taken together, our findings indicate that less sedentary behavior due to stand-biased desks may specifically impact selective attention at different developmental periods. Thus, stand-biased desks represent one option of low-cost intervention aimed at promoting healthy behaviors (i.e., low-level physical activity) by decreasing sedentary behaviors, which may promote further physical activity and downstream effects of cognition in this age range. In this way, our findings may help to elucidate more specific cognitive domains that are impacted by physical activity interventions such as stand-biased desks. 

Notably, female students utilizing stand-biased desks exhibited increased attentional performance compared to their male counterparts, which further underscores the specificity of the impact standing-biased desks have on cognition. Sex differences have not previously been explored within studies on standing desks and cognition in school-age children [17,18]. Structural brain differences between males and females during childhood and adolescence [36] emphasize the need to explore how sex differences impact cognitive development as well. This point is further accentuated by research highlighting sex differences in aerobic exercise and brain development [37]. While our findings build on this work and suggest that females may experience increased benefits in selective attention and cognitive control from utilizing standing-biased, more work needs to be conducted to tease apart these sex-based differences.

Stand-biased desks showed no significant improvements in teacher ratings of observed executive function in the classroom. This demonstrates that stand-biased desks do not affect broader observable executive functioning skills across the day while at school, at least according to teacher report. However, it is worth noting that overall BRIEF-II ratings were relatively typical across the entire sample showing little to no variability, reducing the power to detect change. BRIEF-II forms were designed to provide executive functioning profiles to compare across children with clinical developmental disorders [27]. Due to the relatively typical developmental trajectory of our study’s population, it is possible that effects from stand-biased desks were too marginal to fully examine. Alternatively, teacher ratings of executive functioning may be influenced by other factors aside from objective cognitive performance. It is possible that repeated BRIEF-II teacher report is not a valid indicator of behavioral changes over the year, and more direct observation or parent reports may be more beneficial in measuring these subtle behavioral changes from standing-based desks. Future work teasing apart the impact of standing-biased desks on the executive functioning of children with clinical disorders may show significant changes in these areas. 

There are several limitations within our study that require notice. Participants within the study were not randomly assigned to the traditional or standing-biased desk groups; instead, teachers assigned half their class to each condition. While using a within-subject design helps to account for potential biases in group selection and baseline testing in cognitive function, future studies should utilize random assignment between groups to account for confounding factors that may occur from teacher desk assignments. All cognitive testing was conducted at school during the school day; no existing research has examined the effects of standing desks on behaviors outside of the classroom, specifically at home. In this way, the current research neglects to examine the potential impact of standing-based desks on executive functioning throughout the child’s day and is a relative gap within the literature. Further, we did not examine satisfaction with the standing desks or impact on other outcomes such as youth reported mental health or school satisfaction. Further, while stand-biased desks encouraged more activity among participants, not all children fully utilized the standing nature of the desks allowing for a wider array of variance within subjects. Future work should examine how objectively measured physical activity from stand-biased desks mediates this relationship between stand-biased desks and cognition. Additionally, of note, our study examined the impact of stand-biased desks on cognition within a small number of grades (third, fourth, and sixth graders) and only in one school. Due to the impact that school grade played in moderating the effects of stand-biased desks, studies looking at the full range of primary and secondary students across multiple schools would help to clarify the effects of desks on cognitive development.

## 5. Conclusions

Our findings support that standing-biased desks help improve selective attention and cognitive control among third-grade students and females. While there were no significant differences in the teacher reports of executive functioning, stand-biased desks show some cognitive improvements and no detrimental effects on elementary student cognitive functioning. More work should be conducted to look at the effects of stand-biased desks across all grades in a larger multi-school site sample, as stand-biased desks appear to show specificity depending on the stage of childhood cognitive development. Further, more work should be carried out to tease apart sex differences on stand-biased desk performance on cognition. This work highlights the benefits of stand-biased desks on cognitive functioning during the school day in younger elementary-aged youth.

## Figures and Tables

**Figure 1 ijerph-19-05684-f001:**
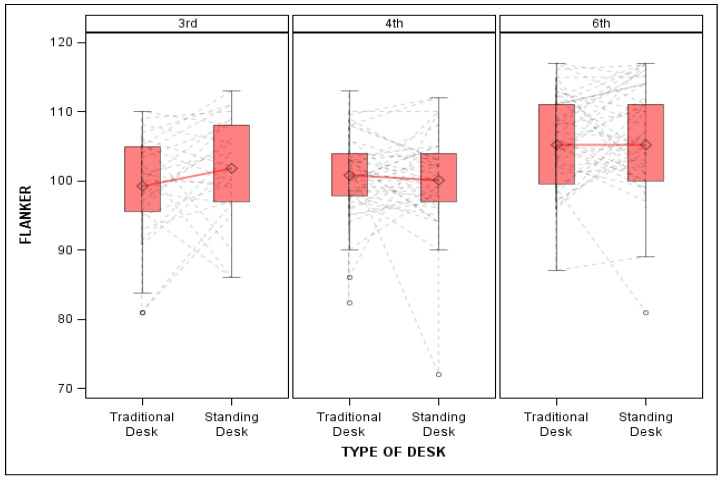
Flanker performance by grade between traditional and stand-biased desks. Figure shows mean flanker performance by grade. Third graders in stand-biased desks showed increased flanker performance compared to their traditional desk using peers.

**Figure 2 ijerph-19-05684-f002:**
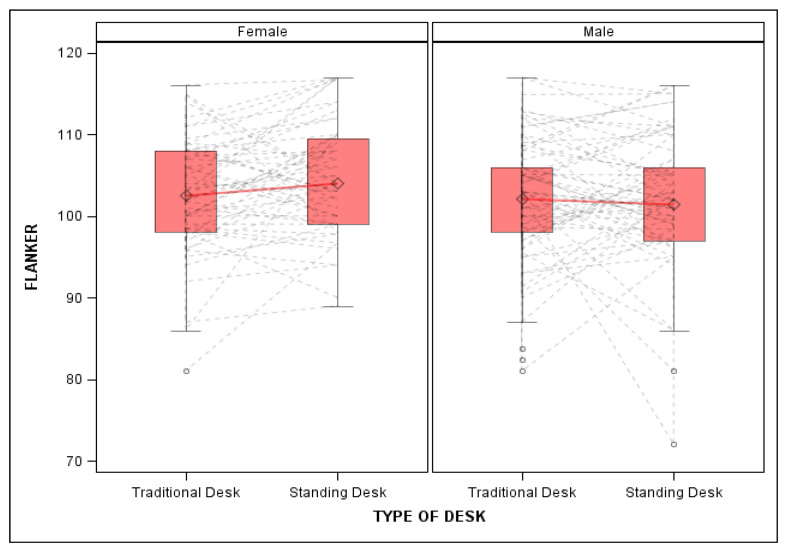
Flanker performance by sex between traditional and stand-biased desks. Figure shows mean flanker performance by sex. Female students in stand-biased desks showed increased flanker performance compared to their traditional desk using peers.

**Table 1 ijerph-19-05684-t001:** Baseline Mean Cognitive and Behavioral Ratings.

	Sit-to-Stand Desk (Group 1) (*n* = 41, 42.27%) M (SD) or %	Stand-to-Sit Desk (Group 2) (*n* = 56, 57.73%) M (SD) or %
Age	10.17 (1.42)	10.29 (1.39)
Female	41.03%	43.10%
Race		
White	75.00%	82.00%
Black/African-America	5.56%	2.00%
Asian	13.89%	6.00%
Mixed Race	5.56%	10.00%
Grade		
3rd	25.64%	18.97%
4th	33.33%	39.66%
6th	41.03%	41.38%

**Table 2 ijerph-19-05684-t002:** Mean Cognitive and Behavioral Ratings Across Time Points.

	Baseline		Time Point 1		Time Point 2	
	Group 1 Sit-to-Stand Desk M (SD)	Group 2 Stand-to-Sit Desk M (SD)	Group 1 Sit-to-Stand Desk M (SD)	Group 2 Stand-to-Sit Desk M (SD)	Group 1 Sit-to-Stand Desk M (SD)	Group 2 Stand-to-Sit Desk M (SD)
NIH Toolbox						
Flanker	101.32 (5.55)	102.97 (7.89)	103.51 (6.77)	104.59 (8.34)	99.97 (6.72)	101.51 (7.70)
List Sort	99.94 (10.86)	100.42 (8.36)	103.06 (11.23)	102.27 (10.72)	102.97 (11.70)	105.51 (8.53)
Pattern Comparison	79.64 (8.28)	81.06 (11.41)	85.68 (7.05)	88.87 (10.05)	99.58 (14.26)	105.10 (16.75)
Picture Sequence	96.54 (10.38)	99.18 (12.90)	101.45 (10.84)	104.98 (12.74)	105.97 (12.88)	107.51 (15.89)
BRIEF-II						
CRI	47.38 (7.55)	48.45 (10.40)	44.79 (5.90)	46.57 (9.79)	44.76 (4.39)	46.28 (9.19)
BRI	48.49 (7.76)	51.13 (13.06)	47.06 (7.57)	47.78 (10.22)	47.33 (8.87)	48.92 (10.62)
ERI	44.81 (5.81)	46.51 (9.96)	45.00 (6.12)	45.54 (9.40)	45.48 (7.31)	45.60 (8.75)
GEC	46.76 (6.80)	48.56 (11.25)	44.85 (5.47)	46.35 (10.04)	45.06 (5.64)	46.44 (9.35)

Note. M = Mean; SD = Standard Deviation; CRI = Cognitive Regulation Index; BRI = Behavioral Regulation Index; ERI = Emotional Regulation Index; GEC = Global Executive Composite.

**Table 3 ijerph-19-05684-t003:** Models of Significance.

	Flanker				Pattern Comparison			
	EST	SE	*p*	f^2^	EST	SE	*p*	f^2^
Intercept	105.44	1.26	<0.001		85.50	2.07	<0.001	
Sequence (REF: Sit–Sit–Stand)	−1.18	1.27	0.353	0.046	−2.95	1.94	0.130	0.088
Grade (REF: 6th Grade)			0.002	0.063			<0.001	0.122
3rd	−6.00	1.68	0.001		−7.71	2.98	0.011	
4th	−4.80	1.49	0.002		−8.64	2.61	0.001	
Female	1.08	1.34	0.418	0.031	2.54	1.97	0.198	0.003
Time (REF: Baseline)			<0.0001	0.157			<0.001	1.976
POST I	1.77	0.76	0.021		5.26	1.96	0.008	
POST II	−1.73	0.69	0.014		25.88	1.87	<0.001	
Standing Desk (REF: Traditional Desk)	−1.91	1.06	0.073	0.066	−1.38	1.25	0.271	0.009
Standing Desk × Grade			0.023					
Standing Desk—3rd Grade	3.65	1.51	0.017					
Standing Desk—4th Grade	−0.41	1.33	0.756					
Standing Desk × Female	2.54	1.18	0.033					
Time × Grade							0.009	
POST I—3rd Grade					3.25	3.07	0.292	
POST I—4th Grade					3.41	2.73	0.213	
POST II—3rd Grade					−3.53	3.05	0.248	
POST II—4th Grade					−6.64	2.66	0.014	

## Data Availability

Not applicable.

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
