# Peer review of "Stand-Biased Desks Impact on Cognition in Elementary Students Using a Within-Classroom Crossover Design"

_ijerph, 2022, doi:10.3390/ijerph19095684_

Round 1
Reviewer 1 Report
This version of the manuscript is clearer, but I still have doubts regarding the Results, particularly, because it has not been well-defined how have the authors incorporated the cross-over design in the ANOVA models for each dependent variable. I also believe authors should report the full ANOVA statistics (e.g., F statistics with the degrees of freedom), including the specific p-value for each ANOVA model tested. I disagree with them that reporting the specific statistics of the analysis will hinder readability. Even if it may, we cannot neglect the rigor of the reporting of the analysis. A balance between readability and rigor of the reporting is paramount in scientific communications.
SPECIFIC COMMENTS
Abstract
Line 55: age or grade? In the Results section the authors don’t present any findings regarding age, only grade and sex…
Introduction
Lines 53–57: These new sentences are a bit confusing, particularly, the excerpt “standing desks have been shown to reduce standing time” (line 55). Can the authors please clarify and amend accordingly?
Lines 79–82: Can the authors simplify the sentence? There’s a lot of “utilizing” hindering readability. At this point, perhaps writing the specific study design is unnecessary.
Line 86: I would add “…during the same period.” at the end of the sentence “…were not exposed to a stand-biased desk”.
Materials and Methods
2.2 Procedures
Please indicate the magnitude of the washout period for the cross-over.
Lines 102–104: There is repeated information. Can the authors fuse these 2 sentences?
“2.3.1 Standing-biased desks.” is Materials and should not be included in the Outcome Measures subheading (only NIH-TB and BRIEF-II)
Line 119: Please insert “later” between “scores” and “utilized”.
When describing the measures/tasks please indicate if a higher or lower score represents a better or worse performance (so that readers can later interpret the tests/tasks Results).
An updated review on the psychometric properties of this BRIEF 2 is available:
Hendrickson, N. K., & McCrimmon, A. W. (2019). Test Review: Behavior Rating Inventory of Executive Function®, Second Edition (BRIEF®2) by Gioia, G. A., Isquith, P. K., Guy, S. C., & Kenworthy, L. Canadian Journal of School Psychology, 34(1), 73–78. https://doi.org/10.1177/0829573518797762
Line 147: Start the sentence with “Baseline”, i.e., “Baseline differences…”. In addition, can the authors comment on how they have used t-test for assessing between-group sex and race differences.
Lines 148–150: Please indicate how was the cross-over design integrated in the ANOVA models and how were the assumptions for conducting the ANOVAs tested.
Results
Please report the full ANOVA statistics (e.g., F statistics with the degrees of freedom), including the specific p-value for each ANOVA model tested. I disagree with the authors that reporting the specific statistics of the analysis will hinder readability. Even if it may, we cannot neglect the rigor of the reporting of the analysis. A balance between readability and rigor of the reporting is paramount in scientific communications.
Table 2: Authors should clarify readers that the “Stand-to- Sit Desk Group” in Time Point 2 was originally the “Sit-to- Stand Desk Group” (as if they would only read Table 2)
Subheadings Grade and Sex
It is not clear whether this analysis includes the cross-over effects
I have not revised the Discussion section because I need to see the Statistical Analysis and Results sections first clarified.
Round 2
Reviewer 1 Report
Dear authors,
This version of the manuscript is much clearer and has higher reporting standards. I have a few minor suggestions:
Lines 100–102: With the respect to later reporting in Results section, e.g., Table 2, perhaps authors can already name Group 1 and Group 2 here.
Lines 118–122: Please remove. This is misplaced and repeated information from previous subheading.
Line 230: The last part of this sentence sounds strange, i.e, “…stages of development compared to than later”. Could it be “… compared to later ones.”, instead?
Best wishes.
Author Response
Please see the attachment.

This manuscript is a resubmission of an earlier submission. The following is a list of the peer review reports and author responses from that submission.
Round 1
Reviewer 1 Report
The theme of this manuscript is interesting. Standing behavior, is a kind of physical activity, even at low levels, may provide both physical and cognitive benefits to children. However, the manuscript still exists many problems.
First, I think the authors need to conduct a sound literature review on the effects of light PA on cognitive or behaviors of children. Standing in the classroom is a behavior of low-level PA. The paper is needed to address the importance of conducting this study.
Second, the study design is confused. Why the authors switch the participants during the intervention. That's nonsense. The authors should follow the rigorous procedures of intervention studies.
Third, why the authors mentioned most participants were White? The paper did not discuss the ethnic difference in using stand-based desk.
Fourth, the manuscript did not mention reliability and validation of instruments used in the study. And why the study chose these instruments.
Based on the above, I think the topic of this study is good. However, there are problems of the study design. The results and conclusion of the study therefore is not reliable.
Author Response
We thank the reviewer for their time. Please see the attachment for our full response.

Reviewer 2 Report
General comments
This is a study analyzing the effects of standing desks versus sitting desks during classes over a school year in cognitive performance measures of third, fourth, and sixth graders. It is an interesting topic of research with potential for contributing to change habits/behaviors at both individual and societal levels. That being said, I find the research design somewhat unclear. The authors appear to be opting initially for cross-over design (lines 93–95) but then they report a pre-post design, analysis, and results, which along with no randomization of students is weakening this research report.
Specific comments
Title
Please add the research design to your Title. Using the PICOS framework is more informative to readers and for researchers conducting systematic reviews of the literature.
Abstract
Please add descriptive and inferential statistical where appropriated.
Introduction
5th paragraph (lines 64–72): This is an interesting reasoning/hypothesis but later this is not studied, is it? Or at least not presented in this report…
Lines 65: Please provide 1 or 2 examples of what is “externalizing problem behaviors” as many readers may not be familiar with this.
Line 66: It’s not clear what the authors are trying to argue, e.g., please provide a direction of association between energy expenditure and externalizing disorders.
Material and Methods
Lines 92–100: Can the authors clarify the research design?
I believe the instruments used in this research to assess cognitive performance (student-based and teacher-based) would be better suite under a subheading “Outcome Measures” instead of “Materials”.
Please provide the psychometric properties, including agreement parameters (e.g., measurement error) and minimal clinical important difference/change (if available) of the instruments used in this research. These, particularly measurement error (SEM, MDC95%) and MCID are very important to interpret the changes over time (natural variation or true effect of interventions).
I find the statistical analysis section overly simplistic. Please provide information regarding alpha level, software used, assumptions, how to interpret effect sizes values, etc.
Results
P values for each variable may be added to the tables.
Overall sample scores pre-post or differences should be presented. Only baseline values are presented.
I stop here because I need to clarification on research design and analysis. Nevertheless, in Discussion section, paragraph 3 (lines 206–221), the intensity of PA in the studies authors are citing were certainly very different from just standing,
usually considered as only 1.5 MET (very light). Please revise.
Author Response

(The authors gave the same response as above.)
